# Progress in Traceable Nanoscale Capacitance Measurements Using Scanning Microwave Microscopy

**DOI:** 10.3390/nano11030820

**Published:** 2021-03-23

**Authors:** François Piquemal, José Morán-Meza, Alexandra Delvallée, Damien Richert, Khaled Kaja

**Affiliations:** Laboratoire National de Métrologie et d’Essais (LNE), 78197 Trappes, France; jose.moran@lne.fr (J.M.-M.); alexandra.delvallee@lne.fr (A.D.); damien.richert@lne.fr (D.R.); khaled.kaja@lne.fr (K.K.)

**Keywords:** calibration method, micro-capacitor, nanoscale capacitance measurements, reference sample, scanning microwave microscopy, uncertainty budget

## Abstract

Reference samples are commonly used for the calibration and quantification of nanoscale electrical measurements of capacitances and dielectric constants in scanning microwave microscopy (SMM) and similar techniques. However, the traceability of these calibration samples is not established. In this work, we present a detailed investigation of most possible error sources that affect the uncertainty of capacitance measurements on the reference calibration samples. We establish a comprehensive uncertainty budget leading to a combined uncertainty of 3% in relative value (uncertainty given at one standard deviation) for capacitances ranging from 0.2 fF to 10 fF. This uncertainty level can be achieved even with the use of unshielded probes. We show that the weights of uncertainty sources vary with the values and dimensions of measured capacitances. Our work offers improvements on the classical calibration methods known in SMM and suggests possible new designs of reference standards for capacitance and dielectric traceable measurements. Experimental measurements are supported by numerical calculations of capacitances to reveal further paths for even higher improvements.

## 1. Introduction

The continuous miniaturisation and integration of components makes the reliable nanoscale characterisation of electrical properties, an absolute critical requirement for successful product development in the electronics industry [1,2,3]. In particular, the accurate measurement of subfemtofarad capacitance at the nanoscale in the gigahertz frequency range constitutes an important challenge for the development of thin films or nanoscale capacitors based on novel dielectric materials with high dielectric constant values for advanced technologies in memory or logic devices [4,5,6,7].

The scanning microwave microscopy (SMM) is a powerful nanoscale technique to measure the admittance of dielectric thin films or nanocapacitors [8,9,10,11,12,13,14,15,16,17]. SMM thus allows the measurement of the nanoscale capacitance and conductance giving access to the dielectric constant, loss angle and dopant concentration of materials. The calibration of these measurements is usually reported in the literature using a classical one-port vector network analyser (VNA) calibration following two main methods developed so far in the literature. The first one is based on a modified short open load (SOL) method using three known capacitance standards [10,11]. The second method is actually derived from electrostatic force sensing based microscopy involving tip-sample approach curves [12,18,19].

To apply or validate the first method, reference materials have been developed from micrometer-sized capacitors [8,9,20] and staircase thin films [12], both using SiO_2_ as dielectric material. In principle, the use of these reference samples provides the traceability of the capacitance measurements to the international system of units (SI) through dimensional measurements. Depending on the calibration method used, the traceability of measurements adopts the known values of the relative permittivity *ε*_r,SiO_2__ for SiO_2_ and the spring constant *k*_AFM_ of the atomic force microscope (AFM) cantilever.

Although important efforts have been made so far related to the calibration methods, a comprehensive uncertainty budget is still missing and several pieces of information remain not accounted for, yet. This mainly concerns some dimensional measurement uncertainty components, repeatability, and uncertainties on key parameters (*ε*_r,SiO_2__ at GHz, *k*_AFM_). At this stage, the range of estimated uncertainties or measurement errors is reported around 10% to 20%.

We have recently demonstrated that a calibrated SMM measurement using a modified SOL method allows the determination of the dielectric constant for SiO_2_ micro-capacitors with a combined standard uncertainty of 15% in relative value [21]. We found that the major contribution to the uncertainty budget was due to the SMM calibration itself.

In the present paper, we show that the uncertainty on the SMM calibration can be significantly reduced to the level of a few percents using improved measurements of dimensional and electrical properties of reference structures. The latter are commercially available calibration kits similar to those used in a previous work [21]. We demonstrate an improved level (same order of magnitude, i.e., a few percents) of combined uncertainties for capacitance measurements in the range 0.1 fF to 10 fF. This improvement relies on comprehensive uncertainty budgets established from investigations of error sources related to the reference structure itself, the instrumentation (SMM unshielded tip) and the environmental conditions (relative humidity).

In the following section, we briefly describe the SMM set-up used in this work, as well as investigated samples and calibration methods. In the next sections, we present a full characterization of the reference standards and results of capacitance measurements using calibrated SMM. The established uncertainty budgets are also discussed further.

## 2. Materials and Methods

### 2.1. Experimental Set-Up

The scanning microwave microscope (SMM) used for the experiments is a Keysight Technologies 5600LS AFM (commercial instruments are identified in this paper for technical clarity and do not imply recommendation or endorsement by the authors) combined with a N5230C Vector Network Analyser (VNA) operating at the frequency of 3.65 GHz. The conductive AFM tip (25Pt300A from Rocky Mountains Nanotechnology) is connected to the RF source/receiver of the VNA through a microwave interferometer based on a Mach-Zehnder configuration [11,15]. The AFM is placed on an active anti-vibration table inside a glove box workstation (MBraun) under a nitrogen atmosphere at room temperature and dry conditions (*RH* < 1%). The whole set-up (glove box, interferometric system and VNA) is installed in a shielded room fitted with a controlled air conditioning system [21].

Technically, the 5600LS AFM is not characterized enough for low uncertainty dimensional measurements [21]. For this, we referred to a well characterized AFM (Nanoman V model from Veeco) for more confident dimensional measurements. Topographic images were thus acquired in tapping mode to avoid any degradation of the sample prior to SMM measurements [22]. For these dimensional measurements, another AFM tip (OTESPA-R3 from Bruker) with smaller apex radius than the conductive AFM tip is used.

In order to carry out traceable 3D topography maps, both AFM instruments were dimensionally calibrated using a LNE-C2N surface topography standard (P900H60) with a pitch of (900.0 ± 1.0) nm and a height of (70.7 ± 0.5) nm [23]. This standard was initially calibrated directly on the LNE metrological AFM [24]. The uncertainties given in this paper are all at one standard deviation corresponding to a 68% confidence level in the case of a normal distribution [25].

### 2.2. Samples

The samples investigated in this work consist of three capacitance calibration kits purchased from MC2 Technologies. Samples are composed of metal oxide semiconductor (MOS) capacitor structures based on SiO_2_ in a variety of configurations.

The MC2 calibration kits, labelled A45, A61, and A64 are composed of 144 identical patterns, each including 48 MOS capacitor structures with capacitance values ranging between 200 aF and 10 fF [9]. These micrometer-sized capacitors are formed by a SiO_2_ dielectric layer sandwiched between circular metallic electrodes (pads) as a top electrode and a highly bore-doped p-type Si (100) substrate as a back electrode. The SiO_2_ layer is structured in 4 plateaus with thicknesses varying from 50 nm to 220 nm by steps of about 50 nm. The metallic pads, 12 per plateau, are formed by a truncated cone composed of a thick Au layer and a thin Ti layer with typically 275 nm overall thickness. Their diameters at the interface with SiO_2_ range between 1 µm and 4 µm, with a 1 µm step. In addition to the 48 MOS capacitors, each pattern presents 2 sets of 4 Schottky diodes consisting in circular gold pads in direct contact to the silicon substrate (no oxide in between) and with diameters ranging from 1 µm to 4 µm [26] (Figure 1).

### 2.3. Calibration Methods

Impedance measurements with SMM require the conversion of the raw measured reflection coefficient *S*_11,m_ data into complex impedances *Z*_s_ of the sample under study. To this end, a modified SOL calibration method is used [10] for traceable impedance measurements. The quantities *S*_11,m_ and *Z*_s_ are related by two equations:(1)$$S_{11}=\frac{Z_{s}-Z_{R}}{Z_{s}+Z_{R}},$$
(2)$$S_{11,m}=e_{00}+e_{01}\left(\frac{S_{11}}{1-e_{11}S_{11}}\right),$$
where *S*_11_ is the theoretical reflection coefficient, *Z*_R_ is the non-zero reference impedance and *e*_00_, *e*_01_, and *e*_11_ are three complex calibration parameters to be determined from *S*_11,m_ values measured on three reference capacitor structures (triplet) with known capacitance values.

The three capacitors used to calibrate the SMM applying the modified SOL method were selected using specific criteria detailed in [21]. In the case of MC2 samples, these reference capacitor triplets (or reference triplet) can be placed at different plateaus of the same pattern.

The sample under study is positioned on the sample holder stage of the SMM at a close proximity to the reference sample used for the calibration procedure. This is with the aim of avoiding large differences in the measurements provoked by the local electromagnetic environment of both samples. The capacitances of the sample under study are calibrated using a so-called substitution method [21]. It consists of substituting the sample under study by the reference sample and then proceeding into measurement cycles as follows:(i)Single topographic and *S*_11,m_ images (phase and magnitude) of the selected capacitor triplet of the reference sample to check the validity of the SMM calibration procedure (i.e., modified SOL method);(ii)Single topographic and *S*_11,m_ images of capacitors on the sample under study;(iii)Single topographic and *S*_11,m_ images of the reference capacitor triplet to check the stability of the SMM calibration.

Performing several sets of these cycles is used to estimate the repeatability uncertainty of the measurements.

Moreover, the data treatment to obtain the three calibration parameters could be done after all the imaging process. The calibration parameters *e*_00_, *e*_01_, and *e*_11_, obtained after the step (i), are applied directly on each *S*_11,m_ images (reference images, sample under study images and control images) resulting in capacitance images.

Since the conductive AFM probes used here are not fully shielded, the measurements of the reflection coefficient *S*_11_ are prone to parasitic and stray capacitances. For this, the *S*_11,m_ images processed using a differential approach in which Δ*S*_11,m_ = *S*_11,m_−*S*’_11,m_ is determined for each scanning line. Δ*S*_11,m_ corresponds to the difference between the raw *S*_11,m_ signals measured on individual capacitors *C*_i_ and the *S*’_11,m_ signals measured on the SiO_2_ layer of the plateau corresponding to the capacitor *C*_i._ This procedure also helps to get rid of the potential scanning drifts occurring during measurements.

## 3. Results

### 3.1. Capacitance Reference Standards

#### 3.1.1. Capacitance Model

• Numerical modeling

Each micro-capacitor structure of the samples is modeled as a circular plate capacitor with a top electrode of a truncated cone shape (finite size radius *R*, height *h*_pad_, angle Θ). This top electrode is separated by a dielectric layer of relative permittivity *ε*_r_ and a thickness *d* from the back electrode considered as an infinite plane, since its equivalent radius is nearly 100 times larger than the largest *R* (2 µm). Its capacitance *C* is calculated by finite element modeling (FEM), using COMSOL Multiphysics 5.3 (AC/DC electrostatic module) to take into consideration the fringing fields [11]. In this work, a 2D axisymmetric model is used, being faster and more accurate than 3D calculations. The top and back plane electrodes of each circular capacitor were set to 1 V and 0 V, respectively. The top, left and right boundaries of the simulation box were set to “zero charge”, i.e., fulfilling the condition $\hat{n}\cdot \vec{D}=0$ where $\hat{n}$ represents the unit vector normal to the boundary and $\vec{D}$ is the electric displacement field. The height *h*_box_ = 100 µm and diameter *D*_box_ = 200 µm of the simulation box are used. The thickness and diameter of the dielectric environment were set to *d* and *D*_box_, respectively. The dielectric constant value *ε*_r,SiO_2__ for the SiO_2_ layer of the capacitors was taken as equal to 3.9 and the global meshing quality was set to “extremely fine” (20 nm). It must be noted here that the best uncertainty on *ε*_r,SiO_2__ obtained so far is in the order of a few parts in 10^3^ [28]. However, this uncertainty has not yet been demonstrated in conjunction with SMM measurements. Hence, we will take into account a conservative uncertainty value of 1%.

• Analytical model

An analytical expression of the capacitance *C* including the terms of *ε*_r_, *R*, *h*_pad_ and *d* is required to determine the uncertainty of the capacitance calculation for the three reference capacitors, and hence the SMM calibration uncertainty. This analytical expression can also be helpful to determine the dielectric constant values of samples under investigation.

When the condition *d* << *R* is fulfilled (it is the case here for most capacitors of the MC2 samples), the fringing fields effects are considered as a correction to the well-known capacitance *C*_p_ of the disk capacitor calculated from the uniform field model:(3)$$C_{p}=\varepsilon_{r\;}\varepsilon_{0\;}\frac{A}{d}\;,$$
with *A* the area of the top electrode and *ε*_0_ the vacuum dielectric constant [29]. The correction due to the fringing fields can be empirically calculated starting from the expression proposed by Sloggett et al. [30], as follows (see Appendix A):(4)$$C_{\mathrm{calc}}=C_{p}\;\left[1+\gamma \left(\varepsilon_{r},h_{\mathrm{pad}}\right)g\left(d,R\right)\right],$$
where γ(*ε*_r,_
*h*_pad_) is an adjustable parameter which depends linearly on *ε*_r_ and *g*(*d*, *R*) given by
(5)$$g\left(d,R\right)=\frac{2d}{\pi R}\left[\ln\left(\frac{8\pi R}{d}\right)-1\right]+{\left[\frac{d}{\pi R}\ln\left(\frac{d}{8\pi R}\right)\right]}^{2}.$$

This expression is more precise than that used in [21], leading to a very good agreement with FEM calculation at the level of a few percentage points for any *R*/*d* values and for *ε*_r_ values ranging from 3.5 to 4.5 (Appendix A).

• Depletion capacitance

Due to a depletion layer occurring inside the doped Si substrate of the back electrode, the capacitance *C*_m_ measured by the SMM includes a depletion capacitance *C*_d_ in series with the capacitance *C* and becomes:(6)$$C_{m}=\frac{C_{d}\cdot C}{C_{d}+C}\;.$$

In the ideal case where the flat band conditions are met at zero voltage applied on the metallic pads, the depletion capacitance *C*_d,FB_ is given by the relation [31]:(7)$$C_{d,\mathrm{FB}\;}=\;\varepsilon_{r,\mathrm{Si}\;}\varepsilon_{0\;}\frac{A}{l_{D}}\;,$$
where *ε*_r,Si_ is the relative permittivity of the highly doped Si substrate (*ε*_r,Si_ = 11.7) and *l*_D_ is the Debye length:(8)$$l_{D}=\;\sqrt{\frac{kT\varepsilon_{r,\mathrm{Si}\;}\varepsilon_{0}}{e^{2}N_{a}}},$$
where *k* denotes the Boltzmann constant, *T* the temperature of the substrate, *e* the elementary charge and *N*_a_ is the doping concentration of the Si substrate. In reality, however, the difference in work function between the metal and the semiconductor results in a local potential drop and leads to an inevitable band bending. In addition the MOS capacitor is affected by charges in the oxide and trapped charges at the interface. The depletion capacitance is then reduced, and in extreme cases, it can reach the minimum value even at zero voltage:(9)$$C_{d,\min\;}=\;\frac{C_{d,\mathrm{FB}}}{2\cdot \sqrt{ln\left(\frac{N_{a}}{n_{i}}\right)}},$$
where *n*_i_ is the intrinsic carrier concentration of Si substrate (*n*_i_ = 1.45 × 10^10^ cm^−3^).

The choice of the reference sample with highly doped silicon substrate makes the *C*_d_ capacitance contribution small but not negligible compared to the capacitance *C* of the dielectric layer. As first approximation, the relative error due to *C*_d_ can be estimated from the mean value of the ratios *C*/*C*_d,FB_ and *C*/*C*_d,min_. In case of a very high *N*_a_ value (about 8 × 10^18^ atoms/cm^3^) measured for MC2 samples (Appendix B), the relative contribution of *C*_d_ to *C*_m_ values is then varying between −1.3% and −4.8% for *C*_m_ in the range 200 aF up to 10 fF.

#### 3.1.2. Determination of the Dimensional Parameters

The geometry of the reference capacitors needs to be very well known in order to ensure the traceability of SMM capacitance measurements to the SI with the highest possible accuracy. To this end, precise measurements of SiO_2_ layer thicknesses *d*, area *A* and height *h*_pad_ of the metallic circular pads were performed on micro-capacitors of samples involving AFM techniques in tapping mode. Table 1 summarizes the SiO_2_ thicknesses *d* measured on patterns of the A45, A61, and A64. We note that the A61 sample presents larger SiO_2_ thickness (except for the first thinnest plateau) compared to the A45 and A64 samples.

The metallic pads of MC2 samples have a truncated cone shape. Therefore, the bottom area of contact with the dielectric layer for each pad has been determined as follows. First, area values measured at five intermediate heights (10%, 30%, 50%, 70% and 90%) have been extrapolated at full height (*h*_pad_ = (275.1 ± 1.9) nm). Then, the area value at 0% height has been corrected by considering the cone angle Θ measured by SEM on several pads, Θ = (19.5 ± 1.5)°. This approach to determine the area from the extrapolated value at 100% height allows one to avoid the correction due to the tip profile and thus to its corresponding uncertainty. More precise AFM combined with scanning electron microscope (SEM) measurements have shown that the bottom part of the pads present a circular step made of titanium of larger radius originating from the etching process made in order to ensure the adhesion of the oxide layer (Figure 2). The mean value of the radius differences measured on several pads was found equal to δ*R* = (40.3 ± 3.6) nm. The area values have thus been corrected accordingly.

#### 3.1.3. Uncertainty Budget on the Capacitance Calculation

The capacitance of the micro-capacitor structures on the MC2 samples (A61, A64 and A45) has been calculated based on the measured values of dimensional parameters and the corresponding uncertainties have been estimated. As an example, Table 2 shows the calculated capacitance values *C*_calc_ of few capacitor triplets used to calibrate the SMM. The relative combined standard uncertainties corresponding to the calculation of capacitances of these capacitors are in the range between 1.4% and 2.8%. They result from the root sum square of the uncertainty components summarized in Table 3 by using the relations (3) to (9). These uncertainties are estimated by applying type A and type B evaluation methods [25]. Type A consists in deducing a probability density from the observed distribution of data. The standard deviation is given by the root square of the variance calculated on repeated sets of observations. In contrast, the type B uncertainties are evaluated from an assumed probability density based on some level of confidence that an event occurs. Compared to the previous work [21], the combined uncertainties on area and thickness measurements, respectively *u*_A_ and *u*_d_ were strongly reduced. They are found to be of the same order of magnitude as those estimated for the SiO_2_ dielectric constant.

The uncertainties on the area measurements are composed of the standard uncertainties related to the repeatability, the image resolution, the pitch AFM calibration and the area correction of metallic pads. These uncertainties are reported in Table 4. The measurement repeatability was determined from 4 images under the same environmental conditions using AFM in tapping mode. The area correction is related to the truncated shape of the metallic pads and the Ti layer as previously described.

The combined uncertainty *u*_d_ on the thickness measurements results from two terms: repeatability and height measurement calibration. The repeatability is determined from 4 series of measurements and amounts, respectively to 1.2% for *C*_01_, *C*_05_, *C*_09_, 0.4% for *C*_13_, *C*_17_ and 0.1% for the other capacitors of thicker dielectric layer (220 nm and 250 nm). The second term is related to the calibration of height measurement in AFM and it leads to an identical uncertainty level of 0.7% for any capacitors.

In addition to the dimensional measurement uncertainties and the uncertainty related to the SiO_2_ dielectric constant, two other uncertainty contributions can be considered. The first uncertainty is related to the error originating from the depletion capacitance *C*_d_ and the second is related to errors induced from the modelling of the capacitances. By using a rectangular distribution delimited by the ratios *C*/*C*_d,min_ and *C*/*C*_d,FB_, the estimated uncertainty corresponding to the correction related to the depletion capacitance is ranging from 0.4% to 2.0%. Without correction, the uncertainty (inside brackets in Table 3) being equal to the error value becomes about twice as good.

The second uncertainty contribution from the capacitance modelling has minor impact on the budget. This includes the uncertainty on the analytical expression used for the calculation of capacitance to propagate the uncertainties. This uncertainty is estimated less than 4 parts in 10^4^. The FEM calculation uncertainty due to error sources such as dimensional modelling of the micro-capacitors, the meshing effect and rounding does not exceed parts in 10^4^. In addition, FEM calculations have been performed using another method based on complementary open sources (FEniCS library, Gmsh mesh generator, and ParaView for visualization application). The results show a good agreement with values calculated using COMSOL at a relative uncertainty of 0.1%.

### 3.2. Capacitance Measurements Using Calibrated SMM

#### 3.2.1. *S*_11_ and Capacitance Images

All *S*_11_ measurements carried out on the above-described patterns show 2D scans (magnitude and phase) with very well defined circular structures for each pad (Figure 3). After converting the *S*_11,m_ images into capacitance maps, the values of the different capacitances (and corresponding standard deviations) are obtained from a Gaussian fit applied to the capacitance histogram determined for each capacitors (Figure 3d). The shape of this histogram shows a homogeneous distribution of capacitance values. This validates the application of the conversion procedure of the magnitude and phase of the *S*_11,m_ raw data into actual capacitances measurements.

It is noteworthy that the diameter of each pad differs between the electrical and topography images, since two different types of probe have been used for these two measurements (see Section 2.1).

#### 3.2.2. SMM Calibrations

The *S*_11_ measurements performed on a given sample were firstly converted into capacitance values by calibrating the SMM using the reference capacitance triplet selected from that sample. It is a prerequisite to check the SMM calibration on a sample from which the three reference capacitances are selected prior to measuring the capacitances on an “unknown” sample. The capacitances of the patterns A61F08, A61E07, A64F07, and A45E08 were obtained by calibrating the SMM using the reference triplets listed in Table 1: (1,17,40) both for A61F08 and A61E07, (5,13,48) for A64F07 and (9,13,48) for A45E08.

As shown in Figure 4, a very good agreement is found between the measured *C*_m,i_ and calculated *C*_calc,i_ values for any *C*_i_ capacitors within the uncertainty. For all patterns, no deviation Δ*C*_i_/*C*_calc,i_ = (*C*_m,i_ − *C*_calc,i_)/*C*_calc,i_ exceeds 11% in absolute value over the full capacitance range, most deviations being within ± 5%. All these deviations are not significant with respect to the expanded uncertainty *U* = 2 × *u* where *u* is the combined standard uncertainty (indicated by error bars in Figure 4). The observed dispersions are due to low precision of the dimensional measurements on pads (area, dielectric thickness). This is particularly the case for the pads of 1 µm diameter where deviations exceeding 5% in absolute value correspond to these small diameter pads. The influence of depletion capacitances has no contribution in the dispersion, since the measured capacitance values depend on the calculated values of the three reference capacitors used to calibrate the SMM. For the four patterns, the mean value of deviations is found to be close to 0%, with a standard deviation less than 3%.

The combined standard uncertainty *u* is estimated for each relative difference Δ*C*_i_/*C*_calc,i_ from the root sum square (RSS) of the standard uncertainty *u*_Ccalc,i_ on the calculation of *C*_calc,i_, and the combined standard uncertainty *u*_Cm,i_ of the *C*_i_ measurements. *u*_Cm,i_ is equal to the RSS of the type A uncertainties *u*_A,i_ (combining *u*_histo,i_ and *u*_rep,i_, which correspond to the standard deviation of the histogram Gaussian distribution of the capacitance *C*_i_ and to the repeatability of the SMM calibration, respectively) and the type B uncertainties *u*_B,i_ mainly dominated by the uncertainty *u*_SMM,i_ estimated for the SMM calibration. *u*_SMM,i_ is calculated from the deviation of the dispersion of 27 capacitance values measured on the capacitor *C*_i_, by considering the 3 values *C*_calc,j_, *C*_calc,j_ + *u*_Ccalc,j_, and *C*_calc,j_ − *u*_Ccalc,j_ for each reference capacitors used to calibrate the SMM [21]. The repeatability uncertainty *u*_rep,i_ was estimated from three series of measurements. For example, Table 5 summarizes the uncertainties on the measured capacitance values for 3 typical capacitors of the pattern A61F08.

The other type B uncertainty components are composed of the uncertainty due to the residual influence of stray capacitances and due to relative humidity as presented below.

• Stray capacitances

The errors due to stray capacitances are strongly reduced in the differential approach (Δ*S*_11,m_ = *S*_11,m_ − *S*’_11,m_), used here as described in Section 2.3. They are much more reduced in the measurements reported here since the capacitors under study and the capacitors used to calibrate the SMM have similar dimensional properties (pad areas, dielectric thickness). Four types of stray capacitance are usually introduced in the case of capacitance measurements between the AFM tip and a doped silicon substrate. They stem from the chip holding the cantilever *C*_chip_, the cantilever itself *C*_lever_, the tip cone *C*_cone_, and the tip apex *C*_apex_. Here, the chip is fully shielded and does not contribute to stray capacitances. The uncertainties corresponding to errors due to the three remaining stray capacitances were estimated using the analytical expressions given in [32,33,34]. Their values are reported in Appendix C.

• Relative humidity

The relative humidity *RH* is expected to affect capacitance measurements because of the water meniscus existing at the interface between the SMM tip and the sample. This effect has been investigated by performing measurements at *RH* values ranging from 0.9% to 4% on two patterns of the same sample: A64-I05 and A64-G07, the latter being used as reference. Appendix D describes the measurement method developed for this purpose.

We find that capacitance values measured at difference humidity levels present very small deviations from those measured at *RH* = 0.9%. Figure 5 shows this deviation to be about (0.17 ± 0.46)% for measurements at *RH* = 2%, (0.33 ± 0.71)% for *RH* = 2.9% and (0.97 ± 1.31)% for *RH* = 4%. We could then fix a conservative uncertainty of 0.2% for measurements in the normal conditions (*RH* ≤ 0.9%).

#### 3.2.3. Capacitance Comparisons

We report here two comparison studies between capacitance measurements on different patterns of the various MC2 samples. The first comparison is made between the patterns F08 from the A61 sample and E08 from the A45 sample. The second comparison is between the E07 pattern on the A61 sample and the F07 pattern on the A64 sample. In the first comparison, the capacitances are measured on A61F08 pattern by calibrating the SMM with the reference triplet selected from A45E08 pattern, e.g., (a,b,c). These measured values have been compared to the measured values on A61F08 pattern with the SMM calibrated using the same set of capacitors (a,b,c) selected from the A61F08 pattern. The same step is done for the second comparison by using the same set of numbered capacitor triplet to calibrate the SMM using the A61E07 and A64F07 (Figure 6).

For these measurements, all the samples were placed side by side around the A61 sample to avoid variation of stray capacitances between cantilever and the other samples. Cantilever is never capping more than one sample.

The reference triplets used for the SMM calibrations have been chosen with micro-capacitors, for which the relative deviations between measured and calculated capacitance values (shown in Figure 5) are found to be less than 5%.

• Comparison over the full capacitance range

The first way to compare capacitance values between two patterns is to consider a reference triplet that covers the full capacitance range. To this end, the choice of the reference capacitor triplet is made such that one of them is among the largest capacitances on the thinner SiO_2_ layer (plateau 1), the other is among the smallest capacitances on the thicker SiO_2_ layer (plateau 4), and the third capacitance on the SiO_2_ layer of intermediate thickness (plateau 2 or 3). The results show (Figure 7) that for the 10 fF capacitances, a large deviation of 10% is found from the comparison between patterns A61F08 and A45E08. The comparison between the patterns A61E07 and A64F07 show an even larger deviation reaching 20%. These relative deviations vary linearly as a function of the capacitance with a slope that depends on the selected reference triplet. It must be noted that the depletion capacitance values are again not considered here in the capacitance calculation since the p-doped Si substrates of samples A61, A45 and A64 present very close doping concentrations between each other (Appendix B).

• Comparison per plateau

The second way to compare capacitances between the samples is to proceed per plateau. We measure the capacitances of a selected pattern of interest (example, A61F08) on a given plateau by calibrating the SMM, using a same set of reference triplets on two different patterns. One of them being the same pattern of interest itself and the other being from another reference sample. However, for both calibration sets, the same plateau level is chosen. The results show a good agreement within ± 5% in the case of comparison between A61F08 and A45F08 patterns (Figure 8a). The micro-capacitors of the 1st plateau show an exception where significant deviations are observed for most of the micro-capacitors. Nevertheless, the micro-capacitors placed on the 3rd and 4th plateau with capacitances less than 1 fF are somewhat in good agreement in the case of comparison between A61E07 and A64F07 patterns (Figure 8b).

The plateau-dependent deviations, which are found different from the two comparisons, are consistent with the SiO_2_ layer thicknesses of the compared samples (Table 1). Indeed, the SiO_2_ layers are thicker for A61 sample than for A45 and much thicker than those of A64.

Such deviations cannot be explained by a difference of stray capacitances between the AFM tip and the samples due to different SiO_2_ layer thicknesses. The stray capacitances do not create errors in the capacitance comparison at the level of 1% (Appendix C).

By fitting linearly the capacitance values, which correspond to capacitors of same diameters, we find that the observed deviations can be explained by parasitic capacitors *C*’ in series with the SiO_2_ capacitors *C*. Their capacitances *C*’ are given by:(10)$$C^{\prime }=\varepsilon_{0\;}\frac{A}{l}\;,$$
with *A* being the area of SiO_2_ equal to that of the top electrodes and a thickness *l* which differs between samples such that *l*_A45_ − *l*_A61_ = 0.67 nm and *l*_A64_ − *l*_A61_ = 1.95 nm. It must be noted that the fit on data for capacitors of the 1 µm diameter in case of measurements on A61F08 has not been considered because of measurement uncertainties. The presence of native SiO_2_ layers on the samples could cause such parasitic contributions. In these cases, the afore-mentioned thickness differences would be multiplied by the value of the relative permittivity of SiO_2_, 3.9, giving rise to thickness difference *l*_A45_ − *l*_A61_ = 2.61 nm and *l*_A64_ − *l*_A61_ = 7.61 nm.

Taking into consideration these parasitic capacitances to correct the measured and calculated capacitance values, Figure 9 shows a good agreement within ±5% over the full capacitance ranges for the two comparisons. The best agreement is found for the capacitance range (0.25 fF − 2.5 fF) provided by the 4th plateau with respect to the typical combined uncertainties for that range, i.e., in the order of 3% (Table 5). The calculation of the mean value of deviations from the 12 capacitances of the 4th plateau and corresponding standard deviations gives <Δ*C*/*C*_calc_>|_A61F08_ = (−2.0 ± 2.4)% and <Δ*C*/*C*_calc_>|_A61E07_ = (−0.3 ± 1.3)% for the measurements on A61F08 and A61E07 patterns, respectively.

Non-compensated linear variations of the capacitance differences as a function of the calculated capacitances in the case of the comparison between A61E07 and A64F07 remain un-corrected. These deviations which in fact vary only with the gold pad diameters could be due to a slight difference of the depletion capacitances between the two samples.

Indeed, these deviations could result from different concentration of charges in the oxide layer and of trapped charges at the interface. Actually, deviations of about 9% can be expected for the 10 fF capacitances in the eventual extreme case where one of the samples presents a depletion capacitance close to the ideal value of flat band condition (*C*_d,FB_), while the other sample presents a depletion capacitance close to the minimum value (*C*_d,min_). Depletion capacitances could also vary from one pattern to another on the same reference sample. Measurements carried out on five patterns of A64 sample with the SMM calibrated using a same reference triplet from a single pattern show a good agreement between the measured values and the mean values within ±2, except for one pattern for which a deviation of 2.9% is observed (Figure 10). These results tend to imply that depletion capacitances are constant over the sample within a few percentage points.

## 4. Discussion

Quantitative measurements of local electrical properties using AFM techniques usually require the use of reference calibration samples. Although this constitutes a fundamental approach, we demonstrate in this work that the use of reference standard should follow careful procedures to unveil implicit dependencies on various experimental conditions.

In this paper, we present a body of work that emphasizes the critical importance of investigating the calibration characteristics of reference samples. Our work focuses mainly on references used in the measurement of local capacitance values and related dielectric properties. We observe several discrepancies between capacitance measurements made on different reference standards using the same SMM system. We also validate these observations using capacitance modeling through numerical calculations, which reveal additional disagreement between theory and experimental measurements. We thus initiate a somewhat comprehensive investigation of possible sources of measurement errors. We also establish an improved budget of uncertainties on these measurements by developing new approaches to account for the sources of errors not fully considered in the prior literature. These encompass errors and uncertainties related to the dimensional characterization of micro-capacitor structures fabricated on the reference samples. The fabrication process itself is also a source of uncertainties, mainly regarding the multilayer nature of the micro-structures (i.e., metallic electrode, charges in oxide layer, interface charges between oxide and semiconductor substrate, doping level of substrate and depletion region).

The main argument in this work focuses on the weight of the different uncertainties depending on the values of measured capacitances of the reference samples. On one hand, it is shown that small capacitances (*C* < 1 fF) are prone to large uncertainties stemming from errors in dimensional analysis of the micro-structures and from the presence of stray capacitances. On the other hand, the measurement of larger capacitances values (*C* > 1 fF) is more affected by artifacts related to the fabrication of the reference sample compared to the design. More specifically, larger capacitances are sensitive to depletion and parasitic capacitances, both originating from either the native oxide layer (having different thicknesses) or from the interface of the dielectric oxide with the semiconducting substrate.

These findings point us towards interesting improvements that could be made on the fabrication and preparation of the reference calibration samples and the built-in micro-capacitor structures. One obvious suggestion would be the use of a metallic substrate, which would lead to the creation of MIM capacitor structures rather than MOS ones. This helps overcoming all uncertainty sources related to charge accumulation at the interface and avoids eventual local non-uniformities in the doping profiles of semiconducting substrates. This latter condition has been shown to be a highly likely source of the observed discrepancies on the reference samples used in this work.

The choice of the dielectric layer follows naturally as a source of uncertainties. In fact, since the reference samples used in this work (MC2 technologies) are fabricated on a doped Si substrate, the choice of SiO_2_ as an oxide dielectric layer in the micro-capacitor structures is somewhat a trivial choice. Although this choice presents some challenging uncertainties as discussed above, it actually offers other advantages. Namely, that the value of the dielectric constant of SiO_2_ is known with an uncertainty less than 1%, as reported in the literature [28]. Thus, using SiO_2_ helps overcoming the uncertainty on the value of the dielectric constant when it comes to measuring the capacitance of the micro-capacitor structures. It follows then, that if one is to prepare new reference samples for capacitance calibration and makes the choice of another dielectric layer to fit the new reference design (i.e., MIM structure), the weight of the uncertainties on the value of the dielectric constants would increase. This argumentation leaves us with few comprising options to propose, depending on the required uncertainty budget for the application of interest. We can easily imagine that the uncertainty budget of about 3% demonstrated in this paper is good enough for most technological applications. Whenever better than 3% levels of uncertainty are required, then the challenge will rely on ensuring a dielectric material with a traceable value of its dielectric constant. Needless to say, the case of the frequency dependence of this sort of metrological characterization of the intended new reference sample is important.

Another critical improvement to consider for better traceable capacitance measurements on reference samples is to highly accurately characterize the geometrical dimensions of the micro-structures. This point could be improved with the use of metrological AFM as it available at LNE. It could reduce the uncertainty in the measurement of the height of plateaus until 0.5 nm. Nevertheless, diameter and area of the gold pads measurements are always impacted by the AFM tip shape convolution with the object.

Moreover, the role of the AFM probe in the calibration of microwave electrical measurements of capacitances is often under-rated. Although many development efforts have been made to produce electromagnetically shielded AFM probes for this goal, their use remains constrained by either high costs or instrumental non-compatibility with most AFM systems. We succeeded in this work to demonstrate that an uncertainty level, as low as 3%, is actually achievable, even with the normal use of unshielded electrical AFM probes. This constitutes an important advantage in SMM measurements, as it offers higher instrumental flexibility with a very low level of uncertainty. Nonetheless, improving the shape of the conductive AFM probes by reducing the conical geometry remains a very viable option for a further improved budget of uncertainties. This could be done by etching the tip by focus ion beam (FIB). More precisely, it would contribute in reducing the source of errors related to stray capacitances.

Finally, we argue on the role of environmental conditions in the measurements of capacitances. We show experimental evidence that relative humidity influences the uncertainty on very small capacitances (*C* < 100 aF). However, this deserves a throughout investigation of the effect of water meniscus formation on the microstructures and its role in the measurement of low capacitances [35,36]. For this, we orientate our focus in this direction using physical modeling by adapting new numerical approaches such as full wave modeling. This will allow one to account for the magnetic field and scattering losses [37]. It would also make it possible to investigate the influence of the frequency of the microwave signal used for probing the sample.

## Figures and Tables

**Figure 1 nanomaterials-11-00820-f001:**
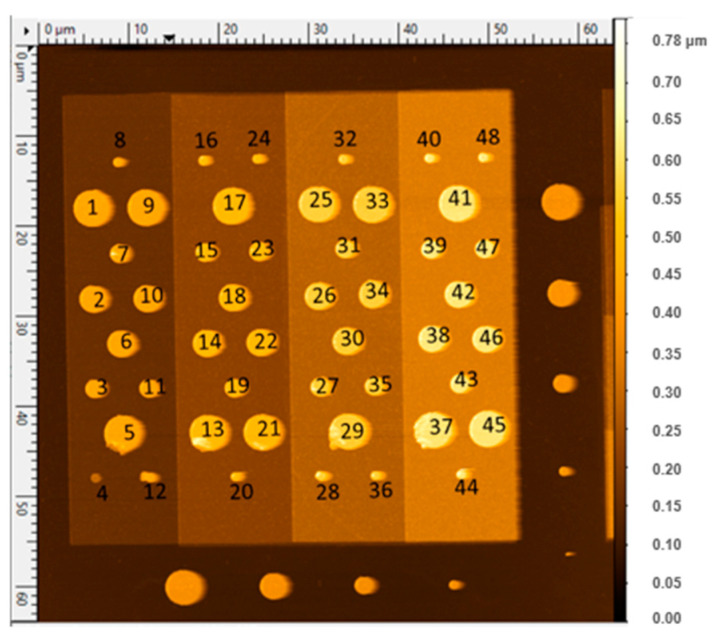
Atomic force microscope (AFM) image of a pattern from A61 sample composed of micro-capacitors labelled 01 to 48 and two sets of Schottky diodes on two sides. This figure has been produced using Gwyddion open source software [27].

**Figure 2 nanomaterials-11-00820-f002:**
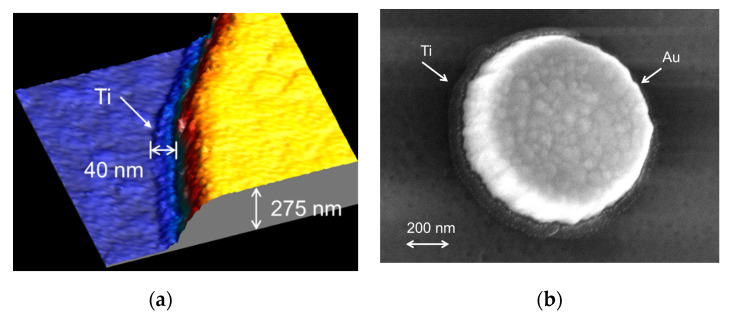
(**a**) AFM image of a gold pad (1 µm diameter and 275 nm height) and an underlying Ti layer with an exceeding 40 nm width on a MC2 sample. (**b**) SEM image of similar gold pad.

**Figure 3 nanomaterials-11-00820-f003:**
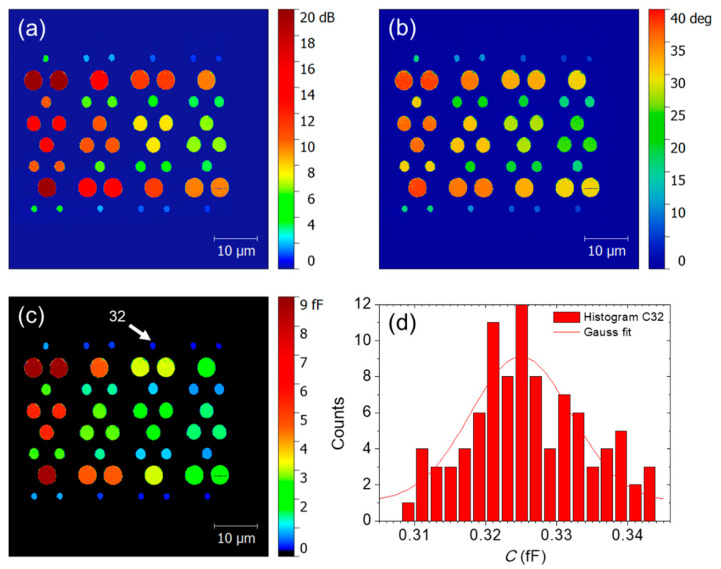
(**a**) *S*_11_ magnitude, (**b**) *S*_11_ phase, and (**c**) capacitance map measured on A61F08 pattern. (**d**) Histogram of capacitance map recorded for the pad 32. The maximum count occurs for a capacitance value which agrees well with the mean value of the Gaussian distribution within the standard deviation of 7 aF.

**Figure 4 nanomaterials-11-00820-f004:**
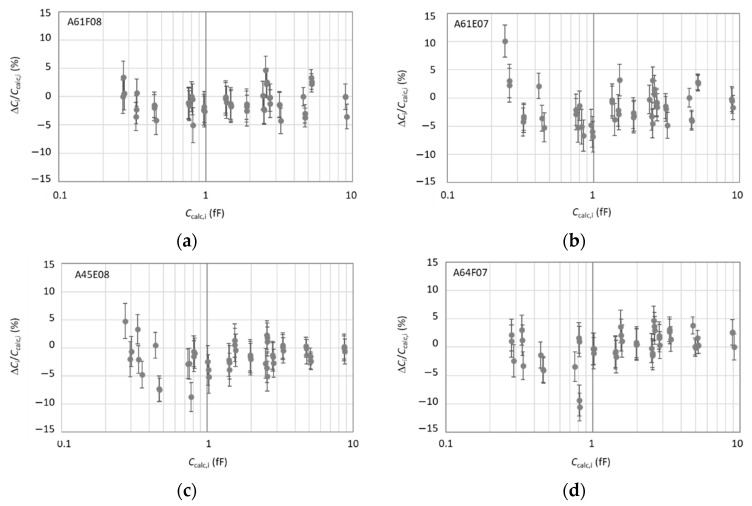
Relative differences Δ*C*_i_/*C*_calc,i_ between measured *C*_m,i_ and calculated *C*_calc,i_ capacitance values for 48 capacitors *C*_i_ of the following patterns: (**a**) A61F08, (**b**) A61E07, (**c**) A45E08, and (**d**) A64F07.

**Figure 5 nanomaterials-11-00820-f005:**
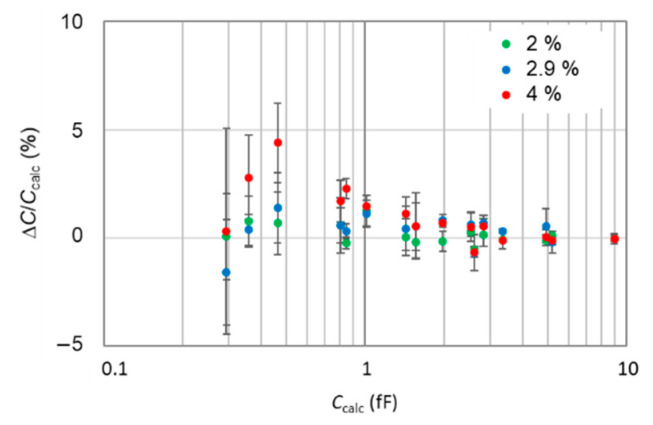
Relative differences between measured capacitances on the pattern A64-I05 at *RH* = 2%, 2.9% and 4% and capacitances measured at *RH* = 0.9%. Each datum represents the mean value of capacitances of 3 capacitors of same area and on same plateau. The error bars correspond to the standard deviation.

**Figure 6 nanomaterials-11-00820-f006:**
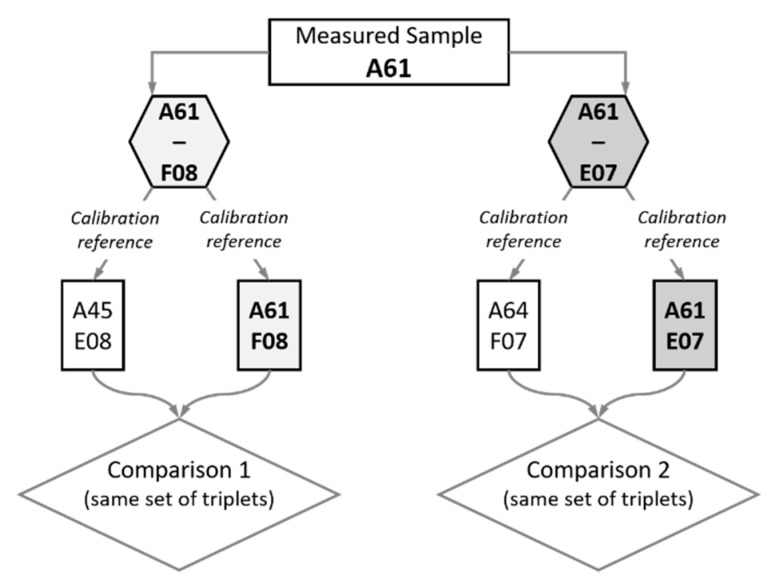
Representative diagram explaining the experimental steps followed in the two-comparison study. The patterns F08 and E07 were selected for this study on the measured sample A61. Measurements of capacitances maps on these two patterns were performed. However, for comparison 1, the SMM has been calibrated using two different reference structures: first A45E08 and then A61F08 itself. These two calibrations were made using the same set of numbered capacitor triplets on each reference structure. A similar workflow has been conducted on the A61 E07 structure, as shown.

**Figure 7 nanomaterials-11-00820-f007:**
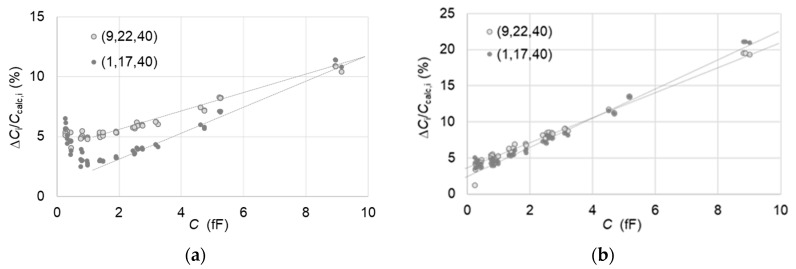
(**a**) Differences Δ*C*_i_/*C*_calc,i_ = (*C*_i,A45_ − *C*_i,A61_)/*C*_calc,i_ in percentage between the measured values *C*_i,A45_ on the 48 capacitors of A61F08 pattern when the SMM is calibrated using the triplet (9,22,40) (full circles) or (1,17,40) (empty circles) from A45E08 pattern and the measured values *C*_i,A61_ when the SMM is calibrated with the same triplets from A61F08. (**b**) Differences Δ*C*_i_/*C*_calc,i_ = (*C*_i,A64_ − *C*_i,A61_)/*C*_calc,i_ with the patterns A61E07 and A64F07 which replace A61F08 and A45E08, respectively. The dashed straight lines are used only to guide eyes. The error bars varying between 2% and 3% have been omitted for clarity.

**Figure 8 nanomaterials-11-00820-f008:**
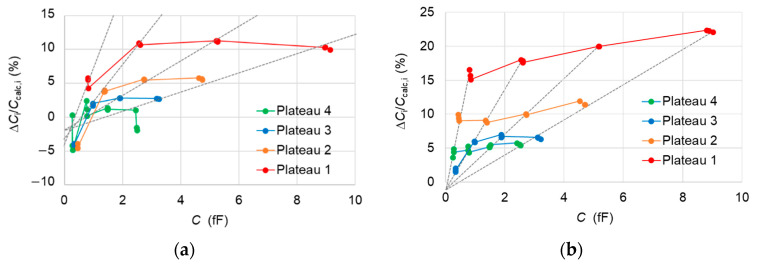
(**a**) Differences Δ*C*_i_/*C*_calc,i_ = (*C*_i,A45_ − *C*_i,A61_)/*C*_calc,i_ in percentage between the measured values *C*_i,A45_ on the 12 capacitors of a given plateau of A61F08 pattern when the SMM is calibrated using triplets from A45E08 pattern and the measured values *C*_i,A61_ when the SMM is calibrated with the equivalent triplets from A61F08. The triplets used are (1,11,12), (17,18,19), (28,30,33) and (41,42,44) for measurements on the plateau 1 (red), 2 (orange), 3 (blue) and 4 (green). (**b**) Differences Δ*C*_i_/*C*_calc,i_ = (*C*_i,A64_ − *C*_i,A61_)/*C*_calc,i_ with the patterns A61E07 and A64F07 which replace A61F08 and A45E08, respectively. The triplets used are (5,6,8), (16,18,21), (29,30,32) and (40,42,47) for measurements on the plateau 1 (red), 2 (orange), 3 (blue) and 4 (green). Dashed lines are linear fits for the capacitors of same diameter. The error bars varying between 2% and 3% have been omitted for clarity.

**Figure 9 nanomaterials-11-00820-f009:**
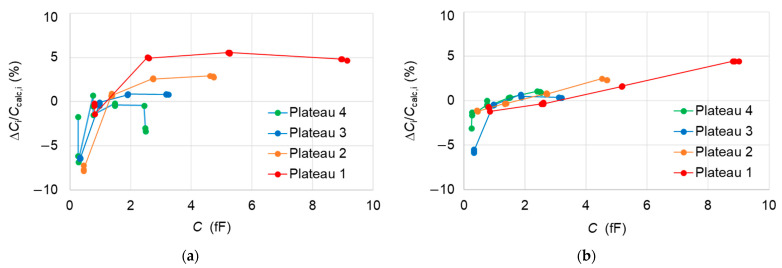
(**a**) Corrected differences Δ*C*_i_/*C*_calc,i_ = (*C*_i,A45_ − *C*_i,A61_)/*C*_calc,i_ in percentage between the measured values *C*_i,A45_ on the 12 capacitors of a given plateau of A61F08 pattern when the SMM is calibrated using triplets from A45E08 pattern and the measured values *C*_i,A61_ when the SMM is calibrated with the equivalent triplets from A61F08. (**b**) Corrected differences Δ*C*_i_/*C*_calc,i_ = (*C*_i,A64_ − *C*_i,A61_)/*C*_calc,i_ with the patterns A61E07 and A64F07 which replace A61F08 and A45E08, respectively. The error bars varying between 2% and 3% have been omitted for clarity.

**Figure 10 nanomaterials-11-00820-f010:**
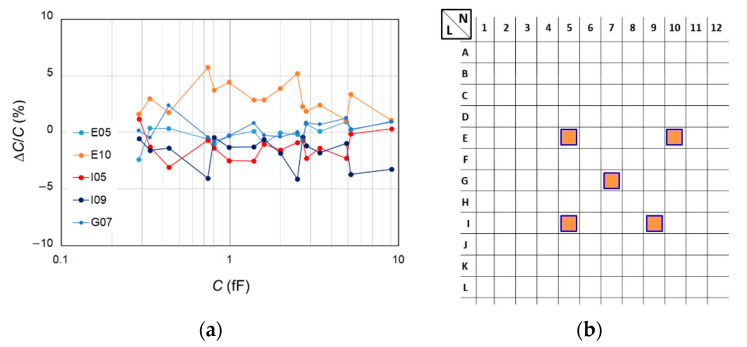
(**a**) Differences Δ*C*/*C* = (<*C*_j,k_> − *C*_j,mean_)/*C*_j,mean_ in percentage between the average <*C*_j,k_> of the measured values from the three capacitors of same type j (same area, same plateau) from the kth pattern (E05, E10, I05, I09, G07) of A64 sample and the mean values *C*_j,mean_ calculated from the 5 patterns. The SMM is calibrated using triplet (05,13,48) from the G07 pattern. (**b**) Localisation of the patterns E05, E10, I05, I09 and G07 inside a zone of 1.28 × 1.60 mm^2^ of the A64 sample. The error bars varying between 2% and 3% have been omitted for clarity.

**Table 1 nanomaterials-11-00820-t001:** Thickness values *d* (nm) and corresponding standard deviations measured for the different plateaus of A45, A61, A64 samples.

	*d* (nm)		
Plateau	A61	A45	A64
1	53.1 ± 0.7	55.4 ± 0.7	53.1 ± 0.7
2	107.1 ± 0.8	105.1 ± 0.8	102.5 ± 0.8
3	162.8 ± 1.2	157.7 ± 1.2	154.2 ± 1.2
4	218.1 ± 1.5	208.6 ± 1.5	204.4 ± 1.5

**Table 2 nanomaterials-11-00820-t002:** Calculated capacitances of reference triplets and combined standard uncertainties. The triplets were selected from patterns labelled A61F08, A64F07, and A45E08.

A61F08		A64F07		A45E08	
Triplet	*C*_calc_ (fF)	Triplet	*C*_calc_ (fF)	Triplet	*C*_calc_ (fF)
*C* _01_	8.94 ± 0.24	*C* _05_	9.16 ± 0.26	*C* _09_	8.68 ± 0.24
*C* _17_	4.62 ± 0.09	*C* _13_	4.96 ± 0.09	*C* _13_	4.78 ± 0.09
*C* _40_	0.27 ± 0.01	*C* _48_	0.29 ± 0.01	*C* _48_	0.30 ± 0.01

**Table 3 nanomaterials-11-00820-t003:** Uncertainty budget corresponding to the capacitance calculation for the capacitors *C*_01_, *C*_05_, *C*_09_, *C*_13_, *C*_17_, *C*_48_ of MC2 samples. Uncertainties are given in relative value (%). Values inside bracket correspond to uncertainty when the error is not corrected.

Uncertainty Budget	Type	*C* _01,_ *C* _05,_ *C* _09_ _(4 µm, plateau 1)_	*C* _13_ *, C* _17_ _(4 µm, plateau 2)_	*C*_40_, *C*_48_ _(1 µm, plateau 4)_
Area measurements, *u*_A_	A, B	0.9	0.9	2.4
Thickness measurements, *u*_d_	A, B	1.3	0.8	0.7
Permittivity *ε*_r_ (SiO_2_), *u_ε_*	B	1.0	1.0	1.0
Depletion capacitance, *u*_Cd_	B	2.0 (4.4)	1.0 (2.3)	0.8 (1.7)
Numerical modelling (COMSOL)	B	0.1	0.1	0.1
Combined uncertainty *u*_C_ (%)		2.7	1.9	2.8

**Table 4 nanomaterials-11-00820-t004:** Relative uncertainties (%) for area measurements on MC2 capacitors.

Uncertainties	*C*_01_, *C*_05_, *C*_09_	*C*_13_, *C*_17_	*C*_40_, *C*_48_
Repeatability (Type A)	0.2	0.2	0.3
Image resolution (Type B)	0.8	0.8	2.3
Pitch AFM calibration (Type B)	0.2	0.2	0.2
Area correction (Type B)	0.2	0.2	0.8
Combined uncertainty *u*_A_ (%)	0.9	0.9	2.4

**Table 5 nanomaterials-11-00820-t005:** Relative uncertainties for 3 typical capacitance values (high, intermediate, low) of the pattern A61F08.

	*C* _05_	*C* _23_	*C* _32_
Capacitance (fF)	(8.79 ± 0.19)	(1.36 ± 0.04)	(0.33 ± 0.01)
Type A uncertainty *u*_A,i_ (%)	0.5	1.3	2.1
*u* _histo,i_	0.1	0.6	2.1
*u* _rep,i_	0.5	1.1	0.2
Type B uncertainty *u*_B,i_ (%)	2.1	2.9	2.6
*u* _SMM,i_	2.1	2.9	2.5
Others	0.2	0.3	0.8
Combined uncertainty *u* (%)	2.2	3.2	3.3

## Data Availability

The data presented in this study are available on request from the corresponding author.

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
