# Peer review of "Progress in Traceable Nanoscale Capacitance Measurements Using Scanning Microwave Microscopy"

_nanomaterials, 2021, doi:10.3390/nano11030820_

Round 1
Reviewer 1 Report
The paper presents a very comprehensive analysis of measurement uncertainties and their sources with use of еру MC2 calibration kits in microwave microscopy. The paper also provides a methodology to account different error sources to reduce the level of uncertainties to several percent. This work will be of interest for those who works with scanning microwave microscopy. The paper is well written and can be published as is. A few minor remarks:
-- the SMM calibration generally can involve two aspects: calibration of the measurement path and calibration of the probe. The calibration dealt with in the presented paper concerns the calibration of the measurement path. Calibration in Ref. 18 is the calibration of the probe. In turn, Ref. 12 clearly distinguishes these aspects, and, apparently, they need to be delineated in the presented paper. In this context, it worth mentioning also the approach by M. Farina et al. in IEEE Trans Microw Theory Tech, 59, p. 2769.
-- few misprints:
82 : thesedimensional
151 : an infinite plan
Author Response
The paper presents a very comprehensive analysis of measurement uncertainties and their sources with use of еру MC2 calibration kits in microwave microscopy. The paper also provides a methodology to account different error sources to reduce the level of uncertainties to several percent. This work will be of interest for those who works with scanning microwave microscopy. The paper is well written and can be published as is. A few minor remarks:
-- the SMM calibration generally can involve two aspects: calibration of the measurement path and calibration of the probe. The calibration dealt with in the presented paper concerns the calibration of the measurement path. Calibration in Ref. 18 is the calibration of the probe. In turn, Ref. 12 clearly distinguishes these aspects, and, apparently, they need to be delineated in the presented paper. In this context, it worth mentioning also the approach by M. Farina et al. in IEEE Trans Microw Theory Tech, 59, p. 2769.
Answer: We thank the reviewer for their comment regarding the delineation between the aspects of the probe calibration and the measurement path calibration. We particularly appreciate the information regarding M. Farina et al which was not known to us previously.
We have changed the sentence (lines 40-42) including reference to Farina’s paper: ‘The second method is actually derived from electrostatic force sensing based microscopy (EFM) involving tip-sample approach curves [12][18][19].’
There are two ways to calibrate capacitance measurements by SMM, the first corresponds to modified SOL method introduced by J Hoffmann [10] and the second proposed both by Gramse and Farina involving a set of three measurements at different tip-sample separation distances. In our opinion, the calibration of the probe alone is not sufficient to insure the SI traceability of the SMM measurements. Nonetheless it’s an integral part of the entire measurement path.
The mentioned works in Ref. 12 and Ref. 18, use a capacitance structure formed by the AFM probe itself as one electrode and the sample’s substrate as a back electrode. Obviously, this makes the contributions of the probe’s geometry directly involved in the correct determination of capacitance measurements. These contributions are dependent on several aspects including the tip-sample separation distance and the size of the probe (tip apex and cone height and opening angle). However, in contrast to these works, our present work here uses the AFM probe merely to mediate an electrical contact with a gold pad that plays the role of one electrode in the micro-capacitor structure. The second electrode being the back electrode on the sample itself. Therefore, the geometry of the gold pad facing the dielectric layer, sandwiched between the pad and the back electrode, has the critical role in defining the uncertainties on the capacitance measurements. An eventual role of the probe geometry here is most likely not excluded, but it has virtually no effect in the analytical procedure followed in our work. This can be explained by taking into consideration the following aspects:
- The probe is in direct contact with the upper gold pad electrode of the micro-capacitor structure. Thus, the probe and the pad are both set to the same electric potential.
- By this fact, there is no additional air gap and distance dependency on the tip geometry that is involved in our measurements.
- Nevertheless, effects related to the probe geometry could be considered from the point of view of overall weighting of stray capacitance created by the probe cone and cantilever with the surrounding gold pads and bare SiO2 surface around the micro-capacitor structure contacted by the probe.
- This overall weighting effect is naturally corrected in the calibration workflow proposed in this work by proceeding into differential measurements. First, the background S11 signal is subtracted for each SiO2 plateau from each image. Second, the measured capacitance is considered with respect to the surrounding bare SiO2 surrounding the gold pad under study.
- These considerations alleviate the contributions of the probe geometry playing a secondary role in our case as overall weighting stray effects.
-- few misprints:
82 : thesedimensional
151 : an infinite plan
Answer: Thank you. The misprints have been corrected.
Reviewer 2 Report
In their Paper "Progress in traceable nanoscale capacitance measurements ...", Piquemal and coworkers report on error sources for capacitance measurements. Such error estimations are for sure of major importance for the development and characterization of nanoscale devices and thus of interest for the readers of nanomaterials. The methodology sounds solid, and the publication continues previous work by the same authors in the field.
However, some questions remain.
On page 7, line 235 the authors introduce "type A and type B evaluation methods". Even so the authors cite a paper from 2008 for that, they should at least give a short description what these evaluation methods are and how they differ.
From Figure 2 it seems that there is quite some roughness present on top of the gold pads. How does this roughness influence the capacitance?
Some minor points:
1. It seems that Figure 1 has been produced using Gwyddion. If so, the authors should give reference to that.
2. Line 151: "plane", not "plan"
3. Figures 7, 8, 9, 10, A1, and B1: The authors should either add error-bars in these plots, or discuss why they don't want to add them.
4. Appendix D: How and where was the temperature measured?
5. The authors should check the correct titles of the given references. Examples:
Line 674: "TE111", not "TE111"
Line 698: "Bell", not "bell"
Because of this, I recommend that the paper should be accepted for publication after a minor revision.
Author Response
In their Paper "Progress in traceable nanoscale capacitance measurements ...", Piquemal and coworkers report on error sources for capacitance measurements. Such error estimations are for sure of major importance for the development and characterization of nanoscale devices and thus of interest for the readers of nanomaterials. The methodology sounds solid, and the publication continues previous work by the same authors in the field.
Answer: We thank the reviewer for their comments.
However, some questions remain.
On page 7, line 235 the authors introduce "type A and type B evaluation methods". Even so the authors cite a paper from 2008 for that, they should at least give a short description what these evaluation methods are and how they differ.
Answer: We have introduced a brief explanation of these two evaluation methods lines 237 to 240.
”Type A consists in deducing a probability density from observed distribution of data. The standard deviation is given by the root square of the variance calculated on repeated sets of observations. In contrast, the type B uncertainties are evaluated from an assumed probability density based on some level of confidence that an event occurs.”
From Figure 2 it seems that there is quite some roughness present on top of the gold pads. How does this roughness influence the capacitance?
Answer: We thank the reviewer for their pertinent remark on the surface roughness of the gold pads. Although it might be tempting to think that this roughness would introduce nanoscale variation of the contact area between the tip apex and the top surface of the gold pad, it is interesting to note that this local contact has practically no effect on the measured capacitances made in this study (see histogram in figure 3). This is because the micro-capacitor structures studied here are actually formed by a dielectric layer sandwiched between an upper gold pad electrode and a lower back electrode substrate. Therefore, it is the lower side of the gold pad, interfacing with the dielectric layer that would play any possible role in varying the measured capacitance of the structure.
Some minor points:
- It seems that Figure 1 has been produced using Gwyddion. If so, the authors should give reference to that.
Answer: Yes, you are right. We have added the sentence “This figure has been produced using Gwyddion open source software [27].” in the figure caption and included the appropriate reference:
- Nečas and P. Klapetek, “Gwyddion: An open-source software for SPM data analysis,” Cent. Eur. J. Phys., vol. 10, no. 1, pp. 181–188, 2012.
- Line 151: "plane", not "plan"
Answer: The misprint has been corrected
- Figures 7, 8, 9, 10, A1, and B1: The authors should either add error-bars in these plots, or discuss why they don't want to add them.
Answer: For figures 7, 8, 9 and 10, the error bars are rather similar for all the data, varying between 2 % and 3 %. They have been omitted for clarity. We have added the sentence “The error bars varying between 2 % and 3 % have been omitted for clarity.” in the figure captions.
For figure A1, the uncertainty for CFEM values is limited to 0.1 %, so too small to be represented in the figure. We have added the sentence “Error bars corresponding to the uncertainty of 0.1 % for CFEM values are too small to be represented.” In the figure caption.
For figure B1, the error bars (rather small) are already represented.
- Appendix D: How and where was the temperature measured?
Answer: Indeed, we have omitted to give this information. The sensor used (Omega) allows us to measure simultaneously the relative humidity, the temperature and the pressure. We have modified the lines 602,603 accordingly:
“The relative humidity is measured using a sensor (Model OM-CP-PRHTEMP101A from Omega TM) placed at the vicinity of the sample. Temperature and pressure have been measured simultaneously using the same sensor.”
- The authors should check the correct titles of the given references. Examples:
Line 674: "TE111", not "TE111"
Line 698: "Bell", not "bell"
Answer: We have checked all the given references, and corrected the references [13], [19],[20], [27], [28], [29], [39]